# Mitochondrial Reprogramming—What Is the Benefit of Hypothermic Oxygenated Perfusion in Liver Transplantation?

**Rebecca Panconesi** [1,2], **Mauricio Flores Carvalho** [1], **Matteo Mueller** [3], **Philipp Dutkowski** [3], **Paolo Muiesan** [1] **and Andrea Schlegel** [1,3,*]

1    Hepatobiliary Unit, Careggi University Hospital, University of Florence, 50134 Florence, Italy;
     rebeccapanconesi@gmail.com (R.P.); drmauras@gmail.com (M.F.C.); paolo.muiesan@unifi.it (P.M.)
2    General Surgery 2U-Liver Transplant Unit, Department of Surgery, A.O.U. Città della Salute e della Scienza di
     Torino, University of Turin, 10124 Turin, Italy
3    Swiss HPB and Transplant Center, Department of Visceral Surgery and Transplantation,
     University Hospital Zurich, 8091 Zurich, Switzerland; matteo.mueller@usz.ch (M.M.);
     philipp.dutkowski@usz.ch (P.D.)
*    Correspondence: schlegel.andrea@outlook.de

**Abstract:** Although machine perfusion is a hot topic today, we are just at the beginning of understanding the underlying mechanisms of protection. Recently, the first randomized controlled trial reported a significant reduction of ischemic cholangiopathies after transplantation of livers donated after circulatory death, provided the grafts were treated with an endischemic hypothermic oxygenated perfusion (HOPE). This approach has been known for more than fifty years, and was initially mainly used to preserve kidneys before implantation. Today there is an increasing interest in this and other dynamic preservation technologies and various centers have tested different approaches in clinical trials and cohort studies. Based on this, there is a need for uniform perfusion settings (perfusion route and duration), and the development of general guidelines regarding the duration of cold storage in context of the overall donor risk is also required to better compare various trial results. This article will highlight how cold perfusion protects organs mechanistically, and target such technical challenges with the perfusion setting. Finally, the options for viability testing during hypothermic perfusion will be discussed.

**Keywords:** liver transplantation; hypothermic oxygenated perfusion; mitochondrial reprogramming; biliary tree

## 1. Introduction

The field of liver transplantation has recently seen a boost in the development of machine perfusion technology with the first randomized controlled trials published and the availability of several devices on the market. To further drive this technology successfully into routine clinical practice, the community needs to first understand underlying mechanisms of protection by different perfusion approaches. Secondly, transparent reports of study results and the identification of gaps in the literature are needed. With this review article we summarize the clinical studies on hypothermic machine perfusion (HMP) in liver transplantation. Next, the protective mechanism of this cold oxygenation is described. Third, a few methodological differences of HMP performed by different groups are identified and discussed followed by the impact of HMP on organ selection through viability assessment. Finally, this review will project future trials needed to answer the here described ambiguities.

## 2. What Is the Real HOPE-Effect in Solid Organ Transplantation?

For a prolonged time, the cascade of ischemia-reperfusion-injury (IRI) was supposed to be initiated by the release of reactive oxygen species (ROS) from phagocytes, and here

mainly macrophages [1,2]. Recently, various mechanistic studies point, however, to an overall and ongoing inflammation, which is seen and pronounced by different cell types, resident in the donor liver and introduced with the recipient blood after normothermic reperfusion. The main instigators of IRI were identified with a high number of mitochondria in hepatocytes (>2000) [3,4]. During warm and cold ischemia, metabolites such as succinate accumulate here. Due to the lack of oxygen, the respiratory chain is unable to maintain the normal electron flow, which leads to a loss of adenosine triphosphate (ATP) and to an accumulation of precursors, including hypoxanthine and xanthine [5]. Another consequence is the accumulation of oxidized nicotine adenine dinucleotide (NADH) at Complex-I. When mitochondria at this stage undergo rewarming and normothermic reperfusion with oxygen, mitochondria aim for a quick reestablishment of their electron flow, however, with incongruent initial electron transfer speeds of the different complex proteins [6–9]. The immediate result is the production and release of reactive oxygen species (ROS) predominantly from Complex-I [9]. Of note this occurs within the first few seconds of warm reoxygenation in various cell types, not exclusively in macrophages (Figure 1) [10–12]. The frequently described tissue inflammation, objectified by various molecules, including cytokines, is considered as a downstream injury and depends on the initial organ quality and the previous duration of ischemia [2,13,14].

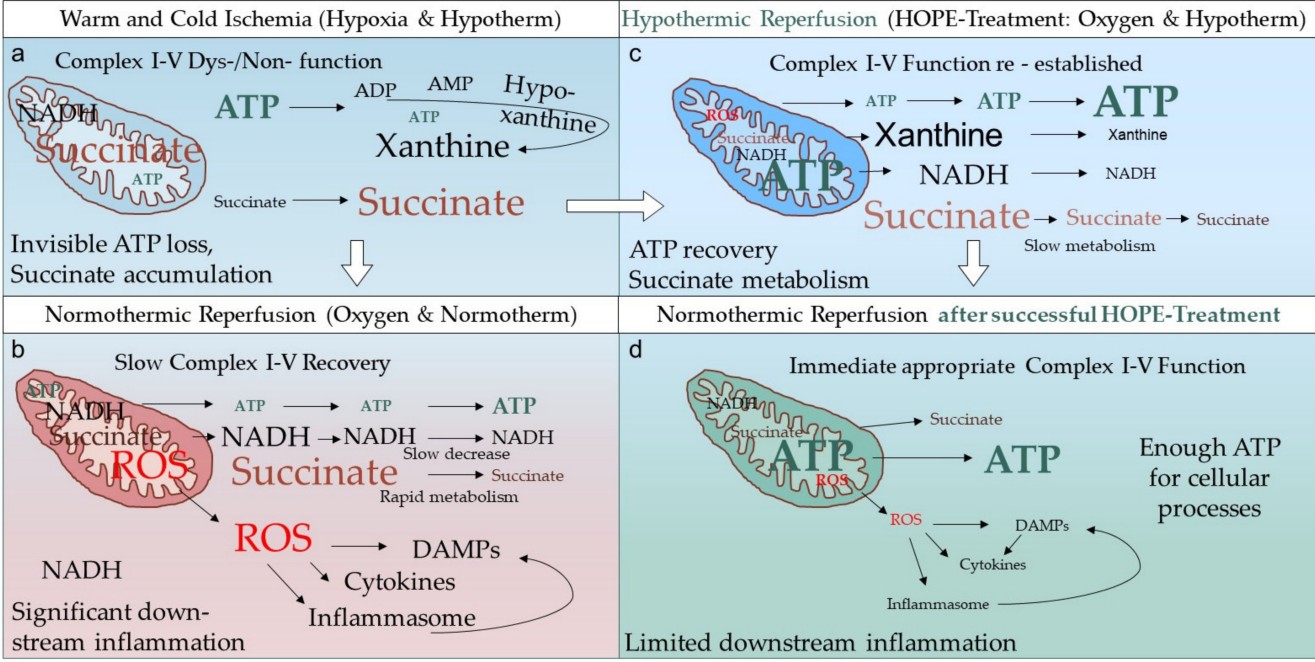

**Figure 1.** HOPE-Effect on mitochondria. During warm and cold ischemia, the tricarboxylic acid cycle (TCA) and the electron transfer is on hold with a subsequent succinate accumulation. Another direct consequence is the loss of ATP, which is broken down to the precursors, hypoxanthine and xanthine [5]. The dysfunctional cellular respiration also leads to a high number of NADH molecules, stuck at Complex-1 (**a**). When such tissues undergo direct normothermic reoxygenation (reperfusion), mitochondria aim to decrease the high load of succinate with the quick reestablishment of the electron flow. This, however, leads to an immediate release of reactive oxygen species (ROS) mainly from Complex-1. Such ROS molecules leak through the mitochondrial membrane and further damage other cell components, with additional release of danger associated molecular patterns (damps) and cytokines. The result of this IRI cascade is an ubiquitarian inflammation, which depends directly on the initial organ quality. Similarly, the time needed to recover cellular ATP and metabolize succinate and NADH after normothermic reperfusion or transplantation is linked to the number of healthy mitochondria able to quickly recover from the IRI (**b**). In contrast, during hypothermic oxygenated perfusion (HOPE, D-HOPE, HMP-O$_2$) the aerobe respiration is reestablished with a normally directed electron flow, very limited ROS release from Complex-1 and full ATP recovery (**c**). Paralleled by a slow succinate and NADH metabolism, such cold pre-treatment of solid organs provides enough cellular energy (ATP) and prevents later ROS release at rewarming and reperfusion under normothermic conditions (**d**).

What is the role of machine perfusion in this cascade? Most dynamic perfusion techniques reintroduce oxygen at different temperatures, thereby inducing an aerobe metabolism. However, at normothermic temperatures this process is accompanied with ROS release and downstream inflammation, because the accumulated succinate is still present when the respiratory chain restarts the electron transfer [5,15–18]. Therefore, the most protective machine perfusion technique should lead to a slow metabolism of accumulated succinate before rewarming to reprogram the respiratory chain with subsequent ATP-reloading before the cellular energy needs increase to levels required for a normal metabolism at 37 degrees (Figure 1) [19]. A similar metabolic situation was also found in mitochondria of the Siberian wood frog, who is naturally protected to live in cold water without oxygen [20]. Another interesting example occurs with hibernators, who are naturally protected from IRI and the related inflammatory injury at rewarming [3,20].

Hypothermic oxygenated perfusion (HOPE) has been identified as the best perfusion technique to achieve the above-described mitochondrial protection [5,18,21]. Reoxygenation in the cold induces a slow succinate metabolism and recovery of Complex-1 to 5 function with efficient ATP recharging. Additionally, a slow metabolism of NADH at Complex-1 has been identified and is used as viability marker for mitochondrial function [5,22]. When mammalian tissue undergoes rewarming at this "prepared" stage, the normothermic reperfusion appears with significantly less ROS release, better organ function, and reduced downstream inflammation [18,23]. Of note, this mechanism is the same in all solid organs, including livers, kidney, and hearts [5,13,24,25].

Another goal of machine perfusion technology is to provide more confidence when accepting an organ through viability assessment. Liver reperfusion at normothermic temperatures is frequently suggested as advantageous for viability testing due to the "near physiologic" behavior during perfusion with a blood-based perfusate at 37° [26–29]. Dynamic parameters, including lactate clearance, bile flow, and composition are considered helpful in this context [29–32]. However, despite a few case series with normothermic machine perfusion (NMP), none of the markers have been linked to biliary complications and graft survival in a large population yet, and various thresholds for viability parameters exist. One example is perfusate lactate, where four different approaches and cut-offs are found in the literature [29,31,33], and in a good number of studies the considered viability parameters were not tested in a transplant model, lacking clinical confirmation [26,29].

Clinically relevant markers to explore liver quality and predict function should ideally be linked to the subcellular structure, which triggers the IRI cascade in mitochondria. Next to the ROS pocket in Complex-1, another molecule, flavin-mononucleotide (FMN), has been identified [34]. At reaction with oxygen, $FMNH_2$ is released into perfusate and plasma and has been found to predict liver function and outcomes after transplantation during HOPE treatment of high-risk livers [5,34,35]. Of particular interest are the autofluorescence characteristics of this FMN molecule, which enable real-time quantification from perfusates through fluorescence spectroscopy during machine perfusion within a few minutes [5,36].

## 3. What Do We Know from Clinical Studies with Hypothermic Liver Perfusion?

In 2010, Guarrera et al. were the first to apply the HMP approach using a "homemade" device in clinical practice [37]. Five years later, the same group presented a lower rate of biliary complications in extended criteria donor (ECD) livers [38]. Despite the steadily evolving number of clinical studies on HMP before liver transplantation, most available trials include only a fairly small, retrospective, and often single center cohort, with various risk profiles and a focus on DCD livers [38–40]. Summarized in a meta-analysis, HMP was found to reduce early allograft dysfunction (EAD) and biliary complications after transplantation [41]. The recent publication of the first randomized controlled trial (RCT) in February 2021 was therefore an important step. Van Rijn et al. showed a significant reduction of ischemic cholangiopathy (IC) by a 2 h endischemic, dual-HOPE-treatment of human DCD livers [42].

Over the last 2 years, the protective effect on other types of ECD livers was verified. For example, the combination of advanced donor age and macrosteatosis or donor warm ischemia time (dWIT) leads frequently to organ decline. The group from Zurich explored the effect of a short HOPE treatment on macrosteatotic human DCD livers, which showed a better outcome after transplantation compared to cold stored controls. Interestingly, in this translational study, authors demonstrate the metabolization of accumulated succinate during HOPE for the first time [13]. This work was paralleled by the group from Turin in 2019 and 2020. Patrono et al. used a dual-HOPE treatment in selected liver transplant cases with advanced risk, including donor age ($\geq$80 years), macrosteatosis, and/or an expected standard cold storage time of $\geq$10 h [43]. Outcomes in both cohorts (n = 25 and 50 cases) were excellent with lower rates of postreperfusion syndrome and acute kidney injury grades 2–3, as well as early allograft dysfunction [43,44]. Overall, there is an increasing body of evidence that a short, cold oxygenation protects recipients from all sorts of complications and graft loss. Three devices are currently in clinical use for liver HMP (ORS® liver transporter, Chicago, Itasca, IL 60143, USA; Liver Assist device from Organ Assist®, Organ Assist Products B.V., 9723 AZ Groningen, The Netherlands; VitaSmart distributed by Bridge to Life®, Northbrook, IL 60062, USA).

The evolving technology also led to a wider range of graft types perfused with the HOPE technique (Figure 2). Livers obtained from pediatric donors were either perfused directly or adult donor livers underwent ex situ split procedures during HOPE or were perfused prior to implantation after completion of the split [45–47]. Interestingly, experimental data support this clinical approach showing a better liver regeneration in HOPE-treated partial liver grafts [48]. Additionally, with a different perfusion route through a surgically reopened umbilical vein, authors from Groningen reported another case series of successful cold perfusion in 2019 [49].

Based on the evidence of protection from IRI, inflammation and posttransplant complications, various groups have started to combine this hypothermic perfusion approach with other dynamic preservation techniques. Triggered by a 20 min mandatory stand-off period, DCD donor livers are routinely procured using normothermic regional perfusion (NRP) in Italy. Following cold flush and storage, an additional HOPE or dual-HOPE treatment is performed and leads to excellent graft and patient survivals with a low number of complications, despite a median functional donor warm ischemia time (fDWIT) of 40 min (IQR: 20–80) [50]. In the context of a better understanding of liver viability, authors from Essen have introduced a controlled oxygenated rewarming (COR), where livers undergo standard HOPE treatment with subsequent gradual rewarming during continuous perfusion until 37° [51]. Using an artificial oxygen carrier, Van Leeuven et al. have performed the same perfusion technique with implantation of the perfused DCD liver grafts [32,52].

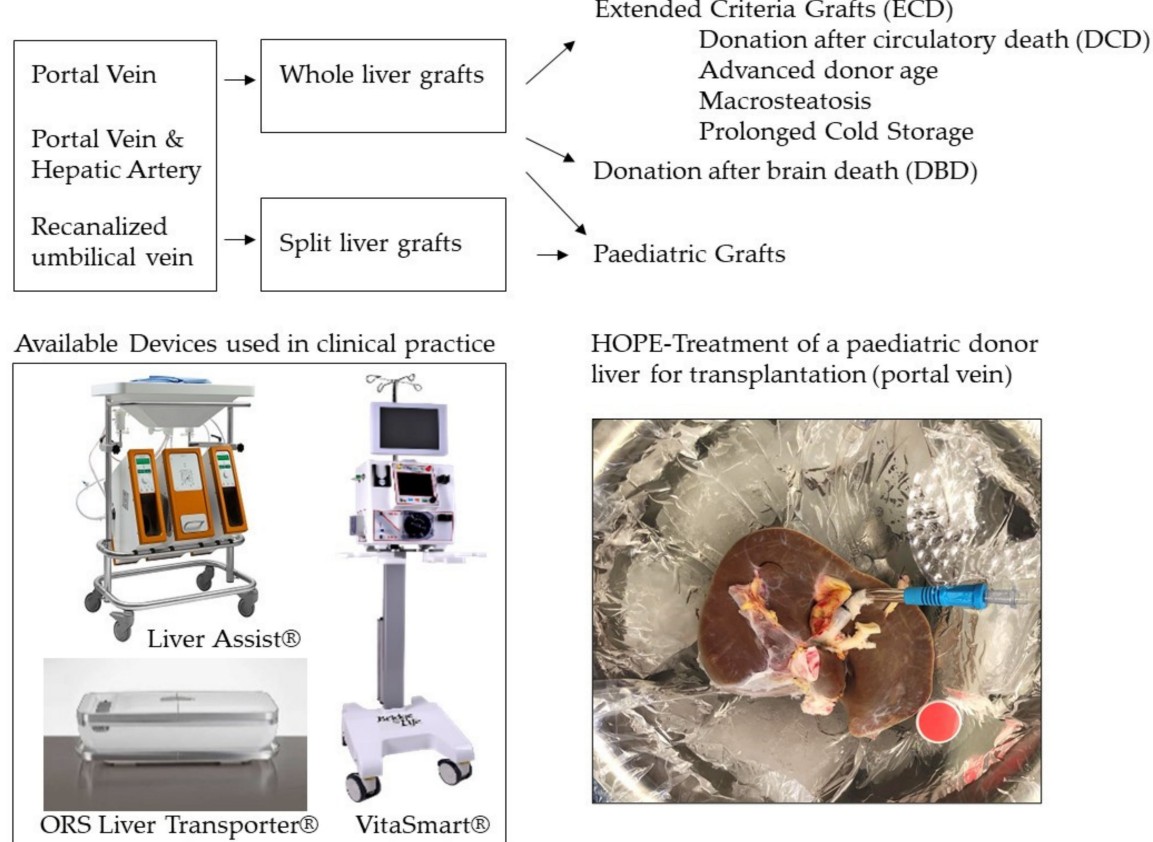

**Figure 2.** Overview on hypothermic liver perfusion in clinical practice. There is clinical evidence for a protective effect of the HOPE concept in various liver types, including ECD grafts and pediatric livers.

## 4. Where Do We Need More Evidence?

Despite such clinical success in the last 5 years, and the better understanding of underlying mechanisms, a few questions remain to be solved. For example, what is the best perfusion protocol, route, and duration? An additional challenge is to determine the range and maximal duration of cold storage prior to HMP, which still leads to the known protective effect with equally good outcomes after liver transplantation. Similarly to NMP, perfusate analysis during HMP revealed more than 2000 molecules, which were quantified to potentially contribute as viability markers [29]. Mitochondrial parameters released from Complex-1 during HOPE were recently found to predict outcomes after implantation of human livers [35]. Subsequently, there is now an increased interest to validate such markers in a large number of human perfusate samples with links to clinical practice. The following section discusses such ambiguities in detail from the background of the available literature.

### 4.1. Do We Need to Perfuse through Portal Vein and Hepatic Artery?

Overall, three main perfusion concepts exist. Guarrera et al. were the first to perform clinical HMP through both inflow vessels (portal vein and hepatic artery) using a self-constructed device [37]. In 2012, the Zurich group translated their simple HOPE approach (through the portal vein only) into clinical practice, using the ECOPS® and later the Liver Assist® device (Organ Assist®) [40]. Derived from this approach and similarly to their colleagues from America, the group from Groningen advocates the additional perfusion through the hepatic artery, as in the dual-HOPE technique [53].

Understandably, the idea of additional perfusion through the hepatic artery was derived from the need of high oxygen pressures for the cholangiocytes in the biliary tree under normothermic conditions. The communication of both inflow vessels is, however, not only known on the sinusoidal level, but the portal vein and hepatic artery also feed both

the vascular mesh around the extrahepatic biliary tree with oxygen [54,55]. This venous and arterial plexus is fed from cranial branches (mainly right hepatic artery and portal vein) and caudal branches (retropancreatic artery and vein) [55]. Based on this anatomy, venous back bleeding from the common bile duct stump is routinely seen following liver reperfusion through the portal vein and before connecting the hepatic artery during transplantation (Figure 3). Mechanistically, the injury of the biliary tree appears two-folded. The protection of the large number of hepatocytes with ATP upload during HOPE leads to a healthier initial bile fluid composition, while the same reprogramming of mitochondria appears in all liver cells, including cholangiocytes and endothelial cells. The HMP approach leads therefore to a reconditioning of cholangiocytes itself (better regeneration after IRI), and also to a better bile composition, because more bile acids are actively (ATP-dependent) secreted from hepatocytes. The cold, oxygenated fluid circulated during HMP reaches the entire biliary tree including the small vascular branches surrounding the extrahepatic bile duct (Figure 3) [56].

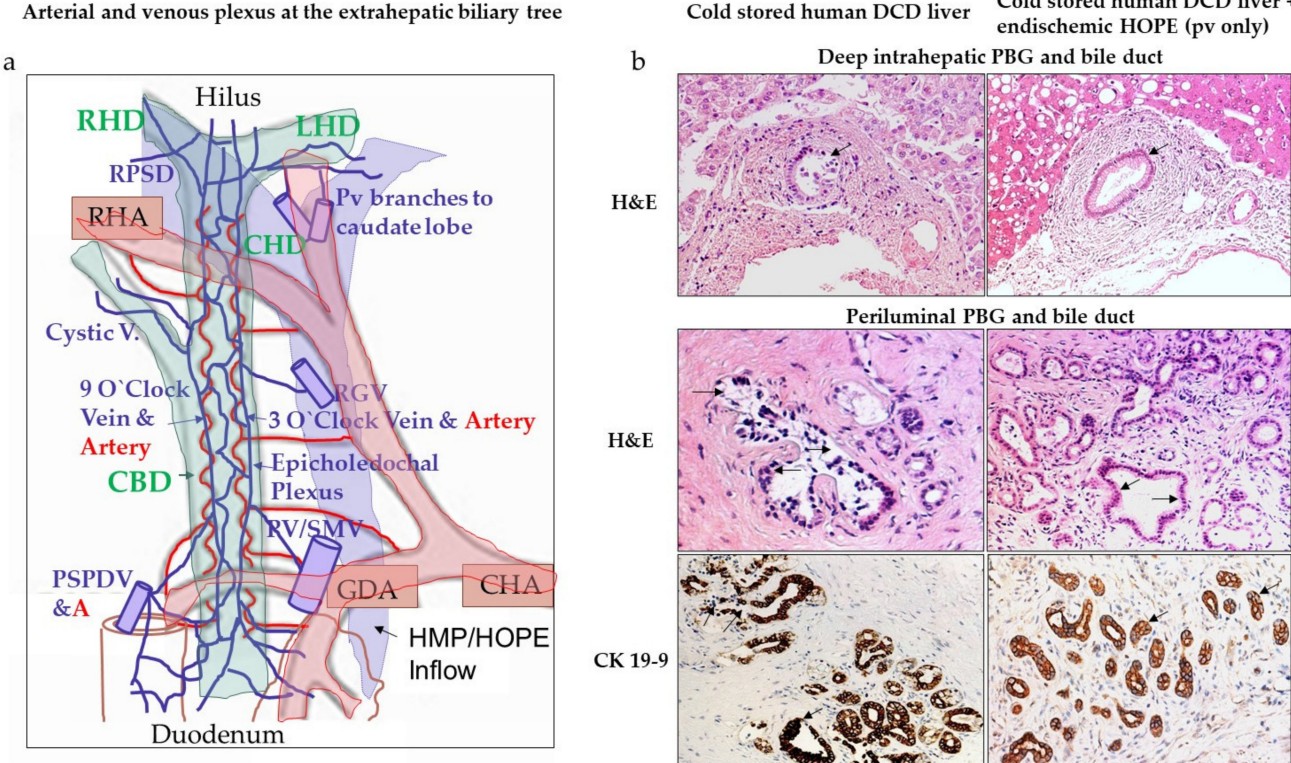

**Figure 3.** Vascular plexus of the extrahepatic biliary tree and histology after transplantation of cold stored and HOPE-treated human DCD liver grafts. Branches from both hepatic artery and venous vessels feed the vascular plexus around the extrahepatic biliary tree, all regional arteries, and veins, including the portal vein (PV) and supra-mesenteric vein (SMV) contribute to the vascular supply (**a**). Histology of deep intrahepatic and periluminal PBG and bile duct in HE and Cytokeratin 19-9 comparing cold stored human DCD livers with HOPE-treated grafts after implantation. HOPE treatment protected the cholangiocellular lining in the ducts by administration of oxygen in the cold fluid through the vascular plexus of the common hepatic and common bile duct (**b**). CBD: common bile duct; CHD: common hepatic duct; CHA: common hepatic artery; GDA: gastroduodenal artery; RGV: right gastric vein; PBG: peribiliary glands; PSPDV + A: posterior pancreatico-duodenal vein and artery.

With such high oxygen partial pressures of 60–80 kPa in HOPE perfusate, enough oxygen molecules are supplied to the entire biliary tree through the portal vein and the vascular plexus. This was further paralleled by an experimental study with administration of fluoresceine through the portal vein during HOPE. Of note, in all three species (rodent, pig, human), the entire liver including the common hepatic duct was stained green within a few

minutes of HOPE, a surrogate for the delivery of oxygen to all liver cells [56]. Importantly, HOPE treatment of high-risk livers has led to superior outcomes after transplantation with regard to the biliary tree, independently of the perfusion route [39,57].

Additionally, in their recent RCT, the Groningen group found 29% and 28% extra-hepatic anastomotic biliary strictures in both arms, despite dual-HOPE (with additional perfusion through the hepatic artery), which is identical to earlier studies with portal vein perfusion only [42,57]. A true comparative trial of both perfusion routes is still lacking. Based on the potential risk of an intima injury through cannulation of the hepatic artery, more evidence is needed to support a general suggestion to perfuse through both vessels in the cold.

### 4.2. Do We Really Need a Transportable Device for Hypothermic Perfusion?

Various technical advances have been made, with three devices available for clinical HMP today [53]. The literature summary, however, clearly shows that this perfusion was always applied after cold storage without device transport (Table 1).

**Table 1.** Overview on clinical studies with hypothermic liver perfusion before liver transplantation. This table summarizes clinical studies with hypothermic liver perfusion (HOPE, D-HOPE, HMP, HMP-O$_2$) within the last 10 years.

| Author and Year | Number and Type of Livers | Duration of Cold Storage before HOPE (min or h) | Duration of HMP (min or h) | Device Transport (Yes/No) | Type of Device | HMP through Portal Vein (PV) or Dually (D) | Main Study Findings |
|---|---|---|---|---|---|---|---|
| Van Rijn et al., 2021, RCT, multicenter | DCD livers, n = 78 | 6 h 11 min (IQR: 5 h 16 min–6 h 55 min) | 2 h 12 min (IQR: 2 h–2 h 33 min) | No | Liver Assist® | D | D-HOPE protects from ischemic cholangiopathy, interventions at the biliary tree and early allograft dysfunction. Same anastomotic stricture rate between D-HOPE group and cold storage control |
| Patrono et al., 2020 ¶ | Extended DBD livers, macrosteatotic, n = 50 | 354 min (IQR: 299–390 min) | 122 min (IQR: 103–176 min) | No | Liver Assist® | D | Low rate of graft loss (2%, n = 1); the level of macrosteatosis correlates with the occurrence of EAD |
| Ravaioli et al., 2019 | Extended DBD liver, donor age, and cold storage (n = 10) | 14.5 h (IQR: 10.8–22 h) | 2.2 h (IQR: 1–3.5 h) | No | VitaSmart® | PV | No PNF and significantly lower rate of EAD, significantly lower recipient transaminases after HOPE treatment and 100% graft survival compared to the cold storage control group |
| Schlegel et al., 2019 * | DCD livers, n = 50 | 4.4 h (IQR: 3.5–5.2 h) | 2.0 h (IQR: 1.6–2.4 h) | No | Liver Assist® | PV | Less PNF, HAT and ischemic cholangiopathy result in a significantly improved five-year survival of HOPE-treated extended DCD liver grafts |
| Patrono et al., 2019 ¶ | Extended DBD livers, macrosteatotic, n = 25 | 311 min ±53 (mean, SD) | 186 min ±49 (mean, SD) | No | Liver Assist® | D | Lower rate of postreperfusion syndrome, acute kidney injury grades 2–3, and lower EAD due to lower recipient transaminases |
| Van Rijn et al., 2018 § | DCD livers, n = 20 | 358 min (IQR: 314–398 min) | 163 min (155–194 min) | No | Liver Assist® | D | D-HOPE treatment reduced reperfusion injury of the biliary tree and led to a better 6- and 12-month graft survival |
| Kron et al., 2018 | Extended DBD (n = 1) and DCD (n = 5) livers, macrosteatotic, n = 6 | 4.7 h | 2.3 h | No | Liver Assist® | PV | HOPE treatment improves immediate liver function and reduces complications and graft loss |
| Van Rijn et al., 2017 § | DCD livers, n = 10 | 395 min (346–457 min) | 126 min (123–135) | No | Liver Assist® | D | Restoration of ATP, lower transaminases and protection of the biliary tree from reperfusion injury and complications through D-HOPE |
| Guarrera et al., 2015 | Extended DBD (n = 31) | 9.3 h ± 1.36 (mean, SD) | 3.8 h ± 0.9 (mean, SD) | No | Own Device | D | HMP showed significantly fewer biliary complications, less EAD and shorter hospital stay |

**Table 1.** *Cont.*

| Author and Year | Number and Type of Livers | Duration of Cold Storage before HOPE (min or h) | Duration of HMP (min or h) | Device Transport (Yes/No) | Type of Device | HMP through Portal Vein (PV) or Dually (D) | Main Study Findings |
|---|---|---|---|---|---|---|---|
| Dutkowski et al., 2015 * | DCD livers, n = 25 | 188 min (141–264 min) | 129 min | No | Liver Assist® | PV | HOPE protected from biliary complications and achieved similar outcomes compared to a matched DBD cohort |
| Dutkowski et al., 2014 * | DCD livers, n = 8 | 141 min | 118 min | No | Liver Assist® | PV | Equal outcomes compared to standard DBD livers, normal immediate function, no biliary complications 8.5 months after LT |
| Guarrera et al., 2010 | Extended DBD (n = 20) | 9.2 h ± 2.1 (mean, SD) | 4.3 h ± 0.9 | No | Own Device | D | HMP showed lower complications, lower EAD rates, and shorter hospital stay |

Studies on combined perfusion approaches or small case series (or reports) with <5 livers were excluded. Study pairs from the same author group are marked with *, §, or ¶ or show a case number, where the smaller cohort is included in the larger one. Numbers are the values as published in the literature, either median or mean, see references. DCD: donation after circulatory death, DBD: donation after brain death, EAD: early allograft dysfunction; ECD: extended criteria donor; D: dual; PV: portal vein; HMP: hypothermic machine perfusion; IC: ischemic cholangiopathy, defined as intrahepatic or hilar strictures of the biliary tree with patent vessels. Cold storage duration is shown in minutes in the original reference was transformed into hours.

Similarly, to hypothermic kidney perfusion, the need for a transportable device is controversially discussed. The majority of ongoing RCTs assess the impact of an endischemic HMP. Studies from Switzerland, Germany, and Italy are expected to be published this year. Ongoing RCTs with HMP worldwide are listed in Table 2. Of note, the American liver RCT is the first study to explore the impact of HMP-O2 after a fairly short period of cold storage using a transportable device [58]. The recruitment is expected to be finished this year.

**Table 2.** Ongoing RCTs on the impact of HMP on outcomes after liver transplantation. The majority finished (or nearly finished) recruitment and results are expected within the next 12–24 months.

| Research Group | Number of Centers | Design | Graft Type | Number of Total Participants | Primary Endpoint |
|---|---|---|---|---|---|
| Zurich (Switzerland) | 13 | Cold storage + HOPE vs. cold storage | DBD (incl. ECD, retransplant) | 170 | Complications 1-year CCI (Clavien III-V) |
| Aachen (Germany) | 4 | Cold storage + HOPE vs. cold storage | DBD (incl. ECD) | 46 | Peak ALT within the first week |
| New Jersey (USA) | 8 | Upfront HMP vs. cold storage | DBD (incl. ECD) | 140 | EAD within the first week |
| Lyon (France) | 8 | Cold storage + HOPE vs. cold storage | DBD (incl. ECD) | 266 | EAD within the first week |
| Bologna (Italy) | 1 | Cold storage + HOPE vs. cold storage | DBD (incl. ECD) | 220 | EAD and DGF within the first 30 days |
| Warsaw (Poland) | 1 | Cold storage + HOPE vs. cold storage | DBD (incl. ECD) | 104 | Model for early graft dysfunction score in first 3 days |
| Bergamo (Italy) | 1 | Cold storage + HOPE with and without cytokine filter | DBD (incl. ECD), DCD | 20 | Postreperfusion syndrome within 5 min after reperfusion |

Additionally, in experimental studies no further ATP increase was found after a prolongation of a cold oxygenation beyond 2 h [59]. Hypothermic liver reconditioning was found with a maximal effect at a two-hour duration. The Groningen group has further established a prolonged dual-HOPE treatment for 24 h, done in the recipient center [60]. The approach of a prolonged preservation is of interest to bridge the time needed to make an intensive care bed available in specific countries, such as the United Kingdom [61]. Based on such literature currently a minimal HOPE treatment of 2 h, which can be safely extended to 24 h, is suggested. However, the need for a transportable device remains

controversial and depends also on regional logistics and the overall donor or liver quality with respect to the cold storage duration. Further mechanistic studies with analysis of the metabolic situation in mitochondria are required to understand what cold storage duration can be accepted prior to HOPE in which donor type.

### 4.3. How Much Cold Storage Time Can We Afford before Hypothermic Liver Perfusion?

The next interesting question appears with the duration of cold storage before the start of HMP. Obviously, a valid answer requires a systematic analysis of reperfusion injury after transplantation of livers with different levels of risk exposed to a range of cold storage durations. Table 1 summarizes clinical studies of HMP published since 2010, where the cold ischemia ranges between 2.35 h (DCD livers) and 14.5 h (extended DBD livers) [40,62]. Although this table includes 11 retrospective and 1 randomized controlled studies with about 265 human livers transplanted with HMP, there is no consensus regarding the suggested duration of cold ischemia before perfusion and general guidelines are lacking.

From kidney transplantation, we learned that the majority of studies explore HMP instead of cold storage, which is in sharp contrast to the field of liver perfusion, possibly because clinicians are afraid of an extended cold ischemia, thereby misinterpreting the cold oxygenated perfusion as ischemia, and naturally, in most studies the cold storage before HOPE is kept rather short [63]. Another cause might be regional factors of the promoting transplant centers for HMP. Travel distances are frequently short, at least initially when a DCD machine perfusion program is started or when centers steadily increase their donor risk to assess the impact of their cold perfusion technique. To date we are not aware of a high-level clinical study clearly demonstrating that prolonged cold perfusion (instead of cold storage) is superior to an endischemic approach, either in kidneys or in liver transplantation [63]. In addition, there is also some mechanistic evidence that an initial cold oxygenation could be superior to the endischemic technique, given the cold storage is prolonged with at least 8 h [24]. In context of the much higher costs with device transport, including the need for more expensive material, manpower, and drivers, a clear comparative study is needed to justify such higher costs for a routine use of HMP in organ transplantation.

### 4.4. Is FMN the Best Mitochondrial Parameter to Predict Liver Function?

Above all, the most important question we need to address targets viability assessment. Mechanistically, the most powerful marker to assess liver function before implantation with the most accurate prediction must be linked to the core of IRI—to mitochondria. Recently FMN was quantified during HOPE perfusion and found to reliably predict outcomes after transplantation [5,29,34–36]. The autofluorescent abilities of this molecule enable the real-time quantification in the operating theater, either through a spectroscope (available at any laboratory) or through novel technology attached to the perfusion tubing, currently under development. Although FMN was found to be a reliable predictor of graft loss beyond the perfusate threshold of >8800 A.U., this will possibly not remain the only molecule which is released from mitochondria during reoxygenation. Currently in use to accept or decline human livers with advanced risk, FMN is measured within 30 min of HOPE treatment. When the perfusate concentration ranges near the cut-off, the measurement will be repeated at the end of the first hour of HOPE [29]. In current clinical practice, the exact FMN value also predicts the specific allocation of a high-risk DCD liver. When FMN appears low (≤5000 A.U.), the graft goes to the initially allocated recipient as planned. Provided that the FMN concentration ranges between 5000 and ≤8800, the liver is reallocated to a healthier recipient, if the primary allocation resulted in a match with a sick, high MELD candidate [29]. Such mitochondrial molecules and the cut-off are currently under international validation with several hundred HOPE-perfusate samples from multiple centers worldwide.

Looking at the metabolic processes in mitochondria during IRI and HOPE, another molecule is considered of additional value, next to FMN. With a normal function of

Complex-I, this molecule is metabolized to the reduced form (NAD+), thereby introducing a proton to establish the gradient for ATP recharging. A high number of NADH molecules are therefore a surrogate of an impaired Complex-1 function and slow recovery of the mitochondrial respiration [5,9,64]. The molecule NADH, with its similar autofluorescent features, can be measured in real time from HOPE perfusates, with a similar fluorescence spectroscopy comparable to the quantification of FMN—just requiring a different wavelength [5,22,29]. In clinical practice, both parameters are considered for liver viability testing with upcoming validation studies.

## 5. Summary and Future Perspectives

Hypothermic oxygenated liver perfusion was successfully introduced in clinical practice. In addition to the recently published RCT, the results of another five randomized trials are currently awaited. In addition, underlying mechanisms of protection were established through reprogramming of mitochondria during HOPE prior to normothermic reperfusion at transplantation. Despite this success, further questions remain with a few technical differences, where the best possible approach with the easiest and most cost-effective device needs to be determined through further clinical and experimental studies. Here the main questions include, first, the duration of cold storage before or after a minimum of two hours of HMP. Next, the best possible perfusion route could for example be clarified with a comparative randomized controlled trial in high-risk DCD livers, where the HOPE treatment is performed with or without perfusion through the hepatic artery (HOPE vs. dual-HOPE). In addition to the two ongoing comparative RCTs between HOPE and NMP, the cold technique should also be compared to NRP in the donor. Although most warm perfusion techniques are frequently discussed as beneficial to administering genes or stem cells, there is also experimental evidence of gene therapy during HMP. This approach should also be explored further to understand potential treatment options to improve mitochondrial function in livers with an FMN release that is currently too high [65].

**Author Contributions:** Conceptualization, R.P. and A.S.; writing—original draft preparation, R.P., M.F.C. and A.S.; writing—review and editing, all authors; visualization: M.F.C. and A.S. All authors have read and agreed to the published version of the manuscript.

**Funding:** The research on hypothermic oxygenated liver perfusion is currently funded by the Swiss National Science Foundation grant no 320030_189055, dedicated to P.D. and A.S. Additionally, P.M. and A.S. are further supported by the University of Florence through grant n° 90-2020/PR.

**Institutional Review Board Statement:** Not applicable.

**Informed Consent Statement:** Not applicable.

**Data Availability Statement:** Not applicable.

**Conflicts of Interest:** The authors declare no conflict of interest. The funders had no role in the design of the review; in the collection of material, or interpretation of data; in the writing of the manuscript, or in the decision to publish the article.

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
