# Peer review of "Mitochondrial Reprogramming—What Is the Benefit of Hypothermic Oxygenated Perfusion in Liver Transplantation?"

_2673-3943, doi:10.3390/transplantology2020015_

Round 1

Reviewer 1 Report

This is a comprehensive and well-written review on the protective mechanisms, current clinical applications, and future directions of HOPE. The authors are highly experienced in this field, and all self-citations are appropriare. I have no particular issues. Compliments to the authors.

Author Response

Reviewer one:

This is a comprehensive and well-written review on the protective mechanisms, current clinical applications, and future directions of HOPE. The authors are highly experienced in this field, and all self-citations are appropriare. I have no particular issues. Compliments to the authors.

Our reply:

We thank the reviewer for the kind evaluation and the comments regarding our manuscript.

Reviewer 2 Report

The present review focus on the benefit of HOPE for liver transplantation. It was well written, and it includes the principal issues related to this topic.

I would suggest adding a section concerning the ongoing randomized clinical studies even not still published and to control the references.

Author Response

Reviewer two:

The present review focus on the benefit of HOPE for liver transplantation. It was well written, and it includes the principal issues related to this topic.

I would suggest adding a section concerning the ongoing randomized clinical studies even not still published and to control the references.

Our reply:

We thank the reviewer for the evaluation of our manuscript and the suggestion We have added a short section and a Table (2) with ongoing RCTs, currently listed under trial.gov. Please see page 8-9 of the revised manuscript. The references were also checked.

Reviewer 3 Report

This is well-organized review article regarding hypothermic oxygenated perfusion in deceased donor liver transplantation.

It will be of help for readers that authors make a table summarizing upcoming RCTs regarding HOPE procedure.

Author Response

Reviewer three:

This is well-organized review article regarding hypothermic oxygenated perfusion in deceased donor liver transplantation.

It will be of help for readers that authors make a table summarizing upcoming RCTs regarding HOPE procedure.

Our reply:

We thank the reviewer for the kind suggestion and have added Table 2 to summarise the awaited RCTs, expected to finish recruitment or to be published. We have also added a few sentences. Please refer to page 8-9 of the revised manuscript.